# Exploring the Role of First-Person Singular Pronouns in Detecting Suicidal Ideation: A Machine Learning Analysis of Clinical Transcripts

**DOI:** 10.3390/bs14030225

**Published:** 2024-03-11

**Authors:** Rong Huang, Siqi Yi, Jie Chen, Kit Ying Chan, Joey Wing Yan Chan, Ngan Yin Chan, Shirley Xin Li, Yun Kwok Wing, Tim Man Ho Li

**Affiliations:** 1Li Chiu Kong Family Sleep Assessment Unit, Department of Psychiatry, The Chinese University of Hong Kong, Hong Kong, China; rong.huang.20@ucl.ac.uk (R.H.); siqi.yi.21@ucl.ac.uk (S.Y.); cjwatson@link.cuhk.edu.hk (J.C.); phoebe.chan@link.cuhk.edu.hk (K.Y.C.); joeywychan@cuhk.edu.hk (J.W.Y.C.); rachel.chan@cuhk.edu.hk (N.Y.C.); ykwing@cuhk.edu.hk (Y.K.W.); 2Division of Psychology and Language Sciences, University College London, London WC1E 6BT, UK; 3Department of Psychiatry, Fujian Medical University Affiliated Fuzhou Neuropsychiatric Hospital, Fuzhou 350000, China; 4Department of Psychology, The University of Hong Kong, Hong Kong, China; shirleyx@hku.hk; 5The State Key Laboratory of Brain and Cognitive Sciences, The University of Hong Kong, Hong Kong, China

**Keywords:** suicidal ideation, clinical interview, first-person singular pronoun, natural language processing, machine learning

## Abstract

Linguistic features, particularly the use of first-person singular pronouns (FPSPs), have been identified as potential indicators of suicidal ideation. Machine learning (ML) and natural language processing (NLP) have shown potential in suicide detection, but their clinical applicability remains underexplored. This study aimed to identify linguistic features associated with suicidal ideation and develop ML models for detection. NLP techniques were applied to clinical interview transcripts (*n* = 319) to extract relevant features, including four cases of FPSP (subjective, objective, dative, and possessive cases) and first-person plural pronouns (FPPPs). Logistic regression analyses were conducted for each linguistic feature, controlling for age, gender, and depression. Gradient boosting, support vector machine, random forest, decision tree, and logistic regression were trained and evaluated. Results indicated that all four cases of FPSPs were associated with depression (*p* < 0.05) but only the use of objective FPSPs was significantly associated with suicidal ideation (*p* = 0.02). Logistic regression and support vector machine models successfully detected suicidal ideation, achieving an area under the curve (AUC) of 0.57 (*p* < 0.05). In conclusion, FPSPs identified during clinical interviews might be a promising indicator of suicidal ideation in Chinese patients. ML algorithms might have the potential to aid clinicians in improving the detection of suicidal ideation in clinical settings.

## 1. Introduction

Suicide is a significant global cause of death, accounting for 1.4% of premature deaths worldwide [1]. Previous research has identified various risk factors associated with suicide, including demographic factors, mental disorders, and hospital visits [2,3]. Mental disorders, in particular, are closely linked to suicide, with a high percentage of individuals who died by suicide having underlying mental health conditions [4,5].

Depression, recognized as a strong predictor of suicide [6,7], is closely tied to self-focus. The self-focus theory proposed by Pyszczynski and Greenberg suggests that excessive self-focused attention plays a role in the development of depression [8]. Research has supported this theory by demonstrating how individual differences in self-focused attention contribute to the risk of depression [9]. Additionally, Durkheim’s social integration theory indicates a correlation between depression and perceiving oneself as detached from society, which further increases the likelihood of suicidal tendencies [10]. Therefore, understanding the development of depression and its connection to suicide requires considering the influence of self-focus.

Language, as a reflection of one’s internal mental state, can provide valuable insights. One linguistic feature associated with self-focused attention and social isolation is the use of first-person pronouns (FPPs), including first-person singular pronouns (FPSPs; e.g., “I”) and first-person plural pronouns (FPPPs; e.g., “we”) [11]. The use of FPPs has been validated as a measure of self-focused attention, showing consistency across different contexts and time [12,13]. Several studies have found a link between the usage of FPSPs and depressive symptoms in both clinical [9,14,15] and non-clinical settings [14,16]. Similar findings have been reported in studies investigating linguistic features of suicide, where the use of FPSPs has emerged as a powerful predictor of suicidal thoughts and behaviors [17].

While the majority of studies have focused on examining FPSPs as a collective entity [17], it is important to note that there exists notable distinctions among specific grammatical categories of these pronouns, including subjective, objective, dative, and possessive cases. Case, as a grammatical category, is determined by the syntactic or semantic function of a pronoun. These varying cases convey unique psychological implications, indicating a research gap in the field [18]. For instance, subjective FPSPs reflect a more active and self-as-actor form of self-focus, while the objective case indicates a more passive and self-as-target form of self-focus [19]. Research on these fine-grained linguistic features remains underexplored and inconclusive, with some studies indicating a significant relationship between the objective case of FPSPs and depression [9], and others suggesting a significant relationship between the subjective case of FPSPs and depression [11], highlighting the need for further investigation.

Recent advancements in artificial intelligence have led to the development of suicide detection systems that utilize machine learning (ML) and natural language processing (NLP) algorithms [20,21,22]. These algorithms analyze textual data from various sources, including social media platforms [23], electronic health records [24], suicide notes [4], and counseling transcripts [25]. ML models have shown promise in distinguishing genuine suicide notes from simulated ones [26,27], detecting suicidal ideation in mental health posts [28], and differentiating users with suicide attempts from controls and users with depression [29]. These approaches have been successful in various cultural contexts, including Asian countries like China [30,31] and Korea [32]. However, to the best of our knowledge, no previous study has specifically investigated the role of different subtypes of FPSPs in the detection of suicidal ideation using transcripts obtained from structured clinical interviews.

This study aimed to explore the usage of different cases of FPPs in transcripts of structured clinical interviews to identify linguistic features that may indicate the presence of suicidal ideation. In addition, this study sought to develop ML models for detecting suicidal ideation, which could potentially assist healthcare professionals in screening for suicide risk. We hypothesized a positive relationship between the use of FPSPs and suicidal ideation, while a negative relationship between the use of FPPPs and suicidal ideation was anticipated, as the latter might imply social inclusion, contrasting the solitary nature associated with FPSPs [11].

## 2. Methods

### 2.1. Participants

This study formed part of an ongoing research project focused on digital phenotyping and the characterization of depression using a case-control design. The inclusion criteria for participation were as follows: (1) being a native Cantonese speaker and (2) being a Chinese adult between the ages of 18 and 65. Participants who (a) had any voice, speech, or language impairments, (b) had a current diagnosis or history of psychiatric disorders other than affective disorders, or (c) were unable to provide informed consent were excluded.

One hundred and ninety-three clinical cases with a lifetime diagnosis of affective disorder (mean age = 53.61 ± 11.77 years; 60% female) were recruited from outpatient clinics in a university-affiliated hospital in Hong Kong and 126 healthy controls without a lifetime diagnosis of affective disorder (mean age = 52.46 ± 11.66 years; 52% female) were recruited from the community between October 2020 and May 2022. In total, this study included 319 participants. The diagnosis of any psychiatric disorder for the clinical cases was determined by the attending psychiatrist and obtained from the review of medical records. The community sample was assessed using the Mini-International Neuropsychiatric Interview (MINI) version 5.0 to identify any DSM-IV diagnoses. Participants were compensated with a cash coupon for their participation. 

### 2.2. Measures

The structured interview guide for the 17-item Hamilton Depression Rating Scale (HDRS) was adopted [33]. All the participants were interviewed and rated by JC, a psychiatrist with MD and PhD degrees [21]. The interviewer did not possess any clinical information regarding the suicidal risk of the interviewees prior to the interview. The overall score of HDRS was used to assess the current depression, with a cutoff score of 8 or above indicating the presence of current depression. H11, which was used to assess suicide risk, asks “Since last week, have you had any thoughts that life is not worth living?” Suicide risk was rated in five progressive levels: (1) having no suicidal thoughts; (2) feeling life is not worth living; (3) having wishes to be dead, or any thoughts of possible death of self; (4) having suicidal ideation or gestures; and (5) having attempts at suicide. The ratings were further validated by TMHL, reaching a kappa of 0.92. The rating of H11 was used to determine suicidal ideation (with a rating of 2 or above as the cut-off point for having suicidal ideation). The interview lasted for around 15–30 min and participants could withdraw from the interview at any time.

### 2.3. Data Preprocessing 

Participants provided verbal responses during the clinical interviews conducted in Cantonese, a colloquial language originating from Guangzhou and the Pearl River Delta region, within the Chinese branch of the Sino-Tibetan language family. The recorded interviews were manually transcribed into Chinese texts by a research assistant with a background in psychology. The transcriptions were then reviewed and verified by TMHL. Once the speech portions of the interviewer were filtered out, the Chinese texts were subjected to text preprocessing using HanLP, an NLP toolkit known for its effectiveness in analyzing texts written in the local language [18]. HanLP provides capabilities such as sentence tokenization, assigning part-of-speech tags to words based on the Chinese Penn Treebank part-of-speech tagset [34], and analyzing the grammatical structure of sentences through dependency parsing, using Stanford Dependencies [35] as a guide. The current study utilized HanLP through its Python implementation, while it is also available in several other languages such as Golang and Java.

With the application of HanLP and other necessary libraries, desired linguistic features, such as FPSPs (and their four subtypes) and FPPPs, along with other common linguistic features including verbs, prepositions, temporal nouns, etcetera, interjections, and passive markers were extracted and tallied automatically [17]. This process involved transforming the textual data into numerical data. Table 1 presents examples of the four subtypes of FPSPs: subjective, objective, dative, and possessive cases. Firstly, in the sentence “我想自殺” (I want to commit suicide), the case of the FPSP “I” was determined as a subjective FPSP due to its role as a nominal subject. Secondly, in the sentence “大家都好憎我” (Everyone hates me), the case of the FPSP “me” was classified as an objective FPSP based on its function as a direct object. Thirdly, in the sentence “佢俾我一個機會” (He gave me a chance), the case of the FPSP “me” was determined as a dative FPSP due to its role as an indirect object. Finally, in the sentence “想自殺係我嘅諗法” (Wanting to commit suicide is my idea), the case of the FPSP “my” was identified as a possessive FPSP, acting as an associative modifier of “idea.” In total, 12 linguistic features (including FPSPs and their four subtypes, FPPPs, verbs, prepositions, temporal nouns, etcetera, interjections, and passive markers) were extracted, and the percentage of their occurrence was calculated by dividing the number of instances of a particular linguistic feature by the total number of tokens identified by HanLP.

### 2.4. Data Analysis

After data preprocessing, logistic regressions were conducted for individual linguistic features as the independent variables, with suicidal ideation as the dependent variable (with two levels: suicidal and non-suicidal). The mean (M), standard deviation (SD), odds ratio (OR) along with its 95% confidence interval (95% CI), and associated *p*-value (with *p* < 0.05 as the threshold of statistical significance) were reported. The logistic regression analyses were adjusted for age, gender, and current depression.

Five commonly used ML models were selected, including gradient boosting, support vector machine, random forest, decision tree, and logistic regression [20]. Five-fold cross-validation was used for testing, that is, the data were equally split into five folds and each time (five iterations in total as there were five folds), one fold was selected as the validation dataset, and the rest were used as the training dataset. For each model, hyper-parameters were tuned by grid search with the aim of achieving the highest and the most balanced specificity and sensitivity by specifying “ROC” as the targeted metric. Moreover, 5-fold cross-validation was included during the model training to make the trained model more robust. The resampling technique, the Synthetic Minority Over-sampling Technique (SMOTE), was additionally employed to address data imbalance in the training dataset. Specifically, during model training, the suicidal ideation rate in the training data was doubled using SMOTE, ensuring a more balanced representation of the target variable. The accuracy, sensitivity, specificity, positive predictive value (PPV), negative predictive value (NPV), F1 score, and area under the curve (AUC) along with its 95% confidence interval (95% CI), and associated *p*-value (with *p* < 0.05 as the threshold of statistical significance) were reported.

## 3. Results

Table 2 illustrates that within the current sample, 38% (120 out of 319) of individuals had current depression and 12% (38 out of 319) of individuals reported experiencing suicidal ideation, with 97.37% of them also being currently depressed. Nevertheless, the statistical analysis conducted did not identify a significant correlation between suicidal ideation and gender (*p* = 0.66) or age (*p* = 0.22).

Table 3 presents the results of the simple logistic regression, highlighting the relationship between each extracted linguistic feature and suicidal ideation. Out of the 12 linguistic features examined, only the use of objective FPSPs demonstrated a significant association with suicidal ideation (OR = 1.20, 95% CI = 2.57–3.47, *p* = 0.02). This implied that for every one unit increase in objective FPSP use, the odds of having suicidal ideation increase by 20%. A similar pattern of result was also observed from the multiple logistic regression, with only the use of objective FPSPs demonstrating a significant association with suicidal ideation (OR = 40.57, 95% CI = 3.84–513.25, *p* = 0.003). Further analyses were conducted to explore the relationship between depression and FPSP use, while controlling for age, gender, and diagnosis of affective disorders. All four cases of FPSPs including subjective (OR = 1.72, 95% CI = 1.34–2.25, *p* < 0.001), objective (OR = 7.44 95% CI = 2.06–28.21, *p* = 0.003), dative (OR = 102.66, 95% CI = 3.03–4892.45, *p* = 0.014), and possessive (OR = 6.44, 95% CI = 2.39–18.18, *p* < 0.001) showed a significant association with depression, whereas FPPPs did not exhibit any significant relationship (*p* = 0.83).

Table 4 presents the evaluation statistics for five ML models that were trained. All models achieved AUCs above 0.5, indicating performance superior to random guessing. Notably, after parameter tuning, the logistic regression and support vector machine (radial) models demonstrated the best performance, with the highest sensitivity and specificity, both achieving an AUC of 0.57 (*p* < 0.05).

## 4. Discussion

To the best of our knowledge, this study represented the first exploration of the relationship between different subtypes of FPSPs and suicidal ideation using transcripts obtained from structured clinical interviews. The inclusion of data from clinical interviews enhanced the relevance and practicality of the findings within clinical settings. Our initial hypothesis proposed a positive relationship between the use of FPSPs and suicidal ideation, as FPSPs reflect self-focused attention, which was known to be associated with depression and an increased risk of suicide. The results provided confirmation of the significance of FPSPs in detecting suicidal ideation, with objective FPSPs emerging as the most influential predictor of suicidal ideation. In addition, we anticipated a negative relationship between the use of FPPPs and suicidal ideation, as FPPPs typically indicated social inclusion and contrast the solitary nature associated with FPSPs. However, no significant associations were found between suicidal ideation and the use of FPPPs in this study. Although FPPPs are generally linked to social inclusion, their absence did not emerge as a significant predictor of suicidal ideation. This finding suggested that other linguistic features (e.g., social words as well as second- and third-person pronouns) or other factors might play a more influential role in indicating social connection or isolation within the context of suicidal ideation.

The limited occurrence of possessive and dative forms of FPSPs in the dataset might have resulted in reduced variability between individuals, thereby limiting the potential associations with suicidal ideation. This also applied to the lack of significant findings for FPPPs, which had a low base rate of 0.02% (which is also why FPPPs were not further divided into different cases). However, despite their low occurrence, the significant finding for objective FPSPs suggested their unique psychological significance. Previous research on depression has suggested that the objective case reflects a more passive, self-as-target form of self-focus. The current findings align with Zimmerman et al.’s study [9], which found that objective FPSPs were a significant predictor of future depression, while subjective FPSPs were not. Considering the close relationship between depression and suicide, the use of objective FPSPs might provide deeper insights into the mental state of individuals at risk. Compared to subjective FPSPs, which reflect a more active and self-as-actor form of self-focus, objective FPSPs highlight the feelings of being a target or a loss of strength in resisting internal challenges [19]. This finding indicated the importance of not only investigating the use of FPSPs but also considering their syntactic position within a sentence. This study, therefore, addressed a research gap by investigating different grammatical cases of FPSPs in suicide detection. It emphasized the need for further exploration of fine-grained linguistic features in suicide-related discourse, including the examination of different grammatical cases of pronouns. Understanding the psychological implications associated with these cases can offer deeper insights into the cognitive and emotional processes related to suicidal ideation.

This study also emphasized the significance of considering cross-cultural differences when investigating language use in the context of mental health. This is particularly relevant because one potential explanation for the absence of a significant association between subjective FPSPs and suicidal ideation could be the phenomenon of pro-drop in Chinese. Pro-drop is observed in languages such as Chinese and Japanese, where the subject of a sentence can be dropped without affecting the sentence’s meaning or grammatical structure [36]. Given that the data analyzed in this study consist of Chinese text, it is highly likely that the prevalence of pro-drop reduced the occurrence of subjective FPSPs, making it challenging to observe a significant effect. Notably, the results indicated a lower frequency of subjective FPSPs compared to studies conducted on English text [11], especially those that reported significant findings. To reconcile these divergent outcomes, future research should delve deeper into the cross-cultural differences in the habitual use of FPSPs and FPPPs.

Through the analysis of linguistic features in clinical interviews, ML models demonstrated potential in aiding healthcare professionals to identify individuals at risk of suicidal ideation, with logistic regression and support vector machine models exhibiting the optimized performance. This suggests the possibility of exploring automated NLP and ML systems for potential support in suicide detection for healthcare professionals. Although it might be argued that the AUC of our ML models (ranged from 0.54 to 0.57) only marginally exceeded the 0.50 threshold of random guess, we believed these results to be fair and promising. It was important to note that we have only utilized a limited number of features in our ML models. It is highly likely that by incorporating more features, the performance of the models can be significantly enhanced. Moreover, the development of Large Language Models (LLMs) has showcased their potential in detecting mental health issues [37]. Thus, the primary objective of our paper was not to develop a comprehensive model, but rather to demonstrate that linguistic features (e.g., FPSPs) might serve as helpful indicators, providing additional information for clinicians as a reference. However, we would not recommend clinicians to solely rely on certain linguistic features to make clinical decisions as we do acknowledge that there are numerous other valuable and influential indicators (e.g., facial expressions). 

It was worth noting that certain prior studies [28,38] have reported higher AUCs, reaching up to 80%, in comparison to the AUCs achieved by our models, which were around 60%. This difference can be attributed to variations in experimental methodologies and data sources. Most previous studies employed a case-control study design, utilizing balanced datasets achieved through equal recruitment of individuals with and without suicidal tendencies or through artificial resampling techniques. While these approaches may yield higher AUCs, their real-world applicability is limited due to the imbalanced prevalence of suicidal ideation in actual populations (approximately 9% [39]). This showed the significance and practicality of our study, as we did not adopt a case-control design or employ extensive resampling techniques (except for the training dataset, but it is still far from balanced). Furthermore, previous studies primarily focused on analyzing social media content, which tends to utilize more explicit language. In contrast, language used during clinical interviews necessitated deeper interpretation and might be influenced by the structure of the interview or the specific questions posed. These factors can potentially constrain the AUCs of our trained models.

This research study provided valuable insights into the predictive role linguistic features might play in identifying suicidal ideation, particularly in cross-cultural and clinical contexts. However, it is crucial to acknowledge the limitations of this study and identify areas for further exploration. One primary limitation is the small sample size, which might have constrained the performance of the ML algorithms employed. To address this limitation, future research should strive to diversify data sources by incorporating information from various clinical contexts, such as psychotherapy sessions and medical consultations. This broader dataset would enable a more comprehensive understanding of how individuals express suicidal ideation. Additionally, with larger datasets available, it would be worthwhile to explore advanced ML techniques, including deep learning, to compare and enhance the performance of automated detection using various techniques. For instance, instead of utilizing NLP techniques, prompts can be used in generative AI to extract features for downstream models. Second, the ML algorithms utilized in this study could benefit from further optimization by training them with more features. The current study primarily focused on linguistic features and did not consider indicators from other modalities (e.g., gestures) or other risk factors associated with suicidal ideation, such as negative events, or hospitalization. Integrating these factors into the analysis could provide a more comprehensive understanding of the complex nature of suicidal ideation.

Another limitation of the current study is its exclusive reliance on clinician ratings of suicidal ideation. While clinician ratings are considered a gold standard in clinical settings, they are subjective and can vary among different clinicians. Future research could consider incorporating objective measures of suicidal behaviors to enhance the objectivity of the data. Moreover, the cross-sectional design of the current study restricted our understanding solely to the mental state of participants on the day of their interview, without insights into any subsequent developments or fluctuations. Such a design does not allow for the establishment of causal relationships, limiting the strength of our findings. Thus, future research might consider a longitudinal study design approach, which allows a more comprehensive understanding of the relationship between linguistic features and suicidal ideation, potentially uncovering causal links or predictive patterns. 

## 5. Conclusions

In conclusion, this study made a valuable contribution to the growing body of research on linguistic markers of suicidal ideation. The findings provided support for the association between the use of FPSPs, specifically objective FPSPs, and suicidal ideation. However, the anticipated negative relationship between FPPPs and suicidal ideation was not observed. This study highlighted the potential of ML models in assessing suicide risk and emphasized the importance of exploring diverse linguistic features and their psychological implications in understanding suicidal ideation. These findings have practical implications for enhancing mental health assessment and provided insights into the potential application of automated NLP and ML systems for detecting suicidal ideation. Further research, using larger and more diverse datasets, is needed to validate and expand upon these findings, taking into account other relevant factors that contribute to suicidal ideation.

## Figures and Tables

**Table 1 behavsci-14-00225-t001:** Examples of data preprocessing.

Subtype of FPSP	Tokenized Text	Dependency Syntax Tree
Subjective case	我 想 自殺 I want suicide ‘I want to commit suicide’	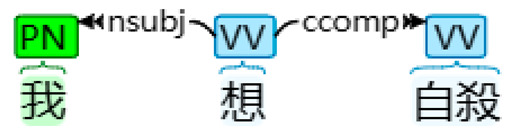
Objective case	大家 都 好 憎 我 Everyone also very hate me ‘Everyone hates me’	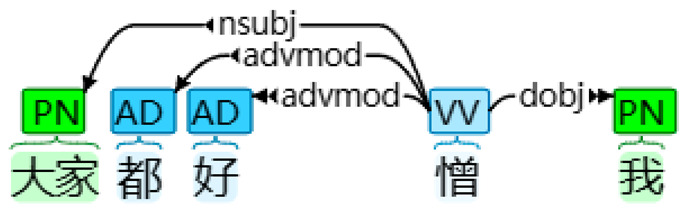
Dative case	佢 俾 我 一個 機會 He give me a chance ‘He gave me a chance’	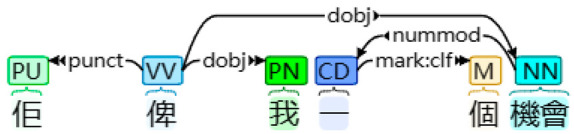
Possessive case	想 自殺 係 我 嘅 諗法 Want suicide is my ‘associative’ idea ‘Wanting to commit suicide is my idea’	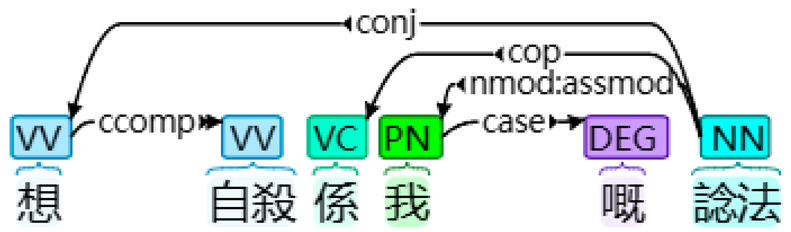

Note. In a dependency syntax tree, boxes display part-of-speech tags, while arrows represent dependencies.

**Table 2 behavsci-14-00225-t002:** Demographics of the participants grouped by suicidal ideation.

Condition	Suicidal (*n* = 38)	Non-Suicidal (*n* = 281)
Age (mean, SD)	50.95 (15.24)	53.46 (11.16)
Gender (n, %)		
Male	15 (39.47%)	122 (43.42%)
Female	23 (60.53%)	159 (56.58%)
Current depression (n, %)		
Yes (HDRS score ≥ 8)	37 (97.37%)	83 (29.54%)
No	1 (2.63%)	198 (70.46%)
Lifetime affective disorder (n, %)		
Yes	34 (89.47%)	159 (56.58%)
No	4 (10.53%)	122 (43.42%)

**Table 3 behavsci-14-00225-t003:** Logistic regression results for 12 linguistic features in predicting suicidal ideation.

Features	Overall Mean (SD)	Non-Suicidal Group Mean (SD)	Suicidal Group Mean (SD)	OR (95% CI)	*p*
Total FPSP	2.23 (1.54)	2.15 (1.52)	2.79 (1.55)	1.00 (0.97, 1.02)	0.91
Possessive (FPSP)	0.25 (0.29)	0.24 (0.29)	0.32 (0.26)	0.98 (0.87, 1.11)	0.77
Subjective (FPSP)	1.78 (1.24)	1.73 (1.23)	2.14 (1.27)	1.00 (0.97, 1.03)	0.94
Objective (FPSP)	0.17 (0.22)	0.15 (0.21)	0.30 (0.25)	1.20 (2.57, 3.47)	0.02 *
Dative (FPSP)	0.03 (0.08)	0.03 (0.08)	0.04 (0.07)	0.72 (0.48, 1.07)	0.10
FPPP	0.02 (0.07)	0.02 (0.07)	0.03 (0.06)	1.51 (0.72, 1.83)	0.55
Verb	20.61 (2.69)	20.49 (2.76)	21.50 (1.82)	1.00 (0.99, 1.02)	0.39
Preposition	0.97 (0.62)	0.96 (0.63)	1.06 (0.48)	1.00 (0.95, 1.05)	0.99
Temporal Noun	0.86 (0.64)	0.86 (0.65)	0.89 (0.56)	0.98 (0.94, 1.03)	0.52
Etcetera	0.02 (0.07)	0.02 (0.07)	0.01 (0.03)	0.77 (0.48, 1.23)	0.28
Interjection	4.52 (1.96)	4.64 (2.00)	3.62 (1.29)	0.99 (0.97, 1.00)	0.21
Passive Marker	0.001 (0.01)	0.001 (0.007)	0.003 (0.02)	2.88 (2.54, 3.45)	0.49

Note. *p*-value < 0.05 was used as the threshold of statistical significance and was denoted with an asterisk (*) symbol.

**Table 4 behavsci-14-00225-t004:** Machine learning evaluation statistics.

Model	AUC (95% CI)	*p*	Accuracy (%)	Sensitivity (%)	Specificity (%)	PPV (%)	NPV (%)	F1 Score (%)
Logistic Regression	0.57 (0.50, 0.64)	0.04 *	64.3	66.6	47.4	90.3	16.1	76.7
Support Vector Machine	0.57 (0.50, 0.63)	0.045 *	64.0	66.2	47.4	90.3	15.9	76.4
Gradient Boosting	0.56 (0.49, 0.62)	0.09	64.3	66.9	44.7	90.9	15.5	75.3
Random Forest	0.55 (0.48, 0.62)	0.13	60.8	62.6	47.4	89.8	14.6	73.7
Decision Tree	0.54 (0.46, 0.61)	0.36	72.4	78.3	29.0	89.1	15.3	83.8

Note. *p*-value < 0.05 was used as the threshold of statistical significance and was denoted with an asterisk (*) symbol.

## Data Availability

The data that support the findings of this study are available from the corresponding author upon reasonable request.

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
