# Peer review of "Exploring the Role of First-Person Singular Pronouns in Detecting Suicidal Ideation: A Machine Learning Analysis of Clinical Transcripts"

_behavsci, 2024, doi:10.3390/bs14030225_

Round 1

Reviewer 1 Report

Comments and Suggestions for Authors

1. Why the control group was from the community (as opposed to hospital outpatients without affective disorder).

2. Did the co-occurrence of COVID-19 pandemic and lockdown during that period have any bearing on study?

3. Was data pre-processing done by the software or humans? Method?

4. What biases were perceived and how the methodology was improved to avoid those biases?

5. How was the distribution of FPSP cases and FPPP for the cohort? What was it for the control group?

6. What is the FPPP equivalent of FPSP Subjective case suicidal ideation? Is it a valid linguistic expression in Cantonese? How do you explain Table-3 p< 0.05 cases, e.g FPSP(sub)?

7. Need explanation of Table-4 model outcome variations.

8. Overall - write-up raises more questions than it answers and hence a thorough rewrite with detailed explanations is needed.

Reviewer 2 Report

Comments and Suggestions for Authors

This study explored the link between linguistic features, particularly first-person singular pronouns (FPSP), and suicidal ideation, using natural language processing (NLP) to analyze clinical interview transcripts. It focused on various forms of FPSP and first-person plural pronouns (FPPP), finding a correlation between the use of objective FPSP and suicidal ideation.

The study focuses on analyzing the association between linguistic features and suicidal ideation, which is interesting. The design of the study and the analysis method look reasonable. There are concerns regarding its results.

Tables 3 and 4 feature multiple comparisons, raising the question of whether p-values were adjusted for this, such as through Bonferroni correction, to mitigate the risk of error from numerous comparisons. This aspect remains unclear.

The Area Under Curve (AUC) value of 0.57, as indicated in Table 4, only marginally exceeds the 0.50 threshold of random guess. This marginal statistical significance casts doubt on the practical utility of these findings in clinical decision-making, as their predictive power is only slightly better than random. The significance and application of these observations in a clinical context are, therefore, questionable.

Additionally, the study does not clarify if interviewers were aware of the subjects' suicide risk, a factor that could potentially introduce bias during the interviews. This uncertainty further complicates the interpretation of the study's findings.

Reviewer 3 Report

Comments and Suggestions for Authors

1.     General comments

The paper addressed the challenge of suicidal detection in clinical applications by investigating the relationship between using linguistic features and suicidal ideation and applying ML techniques to predict suicidal ideation. The methodology and literature review is comprehensive. The case study has shown promising results to utilize the clinical transcripts for better decision making of clinicians and identify the suicide risks more effectively. 

Some suggestions have been provided below to further improve the paper quality:

a.     The prediction from linguistic features may be biased since the suicidal ideation could be hidden in some behavioral features that are not expressed through language. It would be nice to take behavioral factors into consideration to provide a more comprehensive understanding of why linguistic features are important and how the language is reflecting the suicide intention.

b.     From ML perspective, the linguistic features are processed using NLP techniques. With recent development of generative AI and LLM, it would make sense to explore and compare with directly use LLM to process the transcripts and see how it performs. Designing prompts to ask LLM to generate features for downstream models could also be useful.

Round 2

Reviewer 2 Report

Comments and Suggestions for Authors

I remain unconvinced of the model's clinical significance or its usefulness as an indicator, given its AUC value of 0.57. My interpretation of the results leads me to believe that the FPSP in Cantonese might not be an effective tool for assessing suicide risk. Furthermore, I recognize the value of publishing negative findings for the research community, particularly when the study involves a different language or a different population, as is the case with this manuscript. However, I am concerned that the authors may be portraying these negative results as if they were positive. Beyond these points, I have no additional comments on the manuscript.

Author Response

We appreciate your comment and understand your concerns. We fully acknowledge that the clinical significance of our model is limited, as indicated by our suggested directions for future studies. In fact, our team is actively working on exploring these directions.

Although our results may not be deemed as strictly positive findings, we consider them to be fair and promising. Throughout the paper, we have taken a cautious and open approach in interpreting our results. We have used very tentative language, such as "might," "potential," and "possibility," to avoid making strong and definitive claims about the clinical significance of our findings.

While we maintain a cautious stance regarding the current results, we remain optimistic for the future based on the findings presented. It is important to note that the primary objective of this paper is not to develop a powerful model, but rather to emphasize the significance of investigating diverse linguistic features and their psychological implications in understanding suicidal ideation.

Thank you for your valuable feedback, which allows us to clarify our intentions and ensure the accurate representation of our work. Please be assured that we have further refined our language to ensure the appropriate interpretation of our results.